# The Reliability of Carotid Artery Doppler Ultrasonography Indices in Predicting Fluid Responsiveness during Surgery for Geriatric Patients: A Prospective, Observational Study

**DOI:** 10.3390/healthcare12070783

**Published:** 2024-04-03

**Authors:** Beliz Bilgili, Ayten Saracoglu, Kemal T. Saracoglu, Pawel Ratajczyk, Alper Kararmaz

**Affiliations:** 1Department of Anesthesiology and Intensive Care, Marmara University Pendik Training and Research Hospital, 34899 Istanbul, Turkey; beliz.bilgili@marmara.edu.tr (B.B.); alper.kararmaz@marmara.edu.tr (A.K.); 2Department of Anesthesiology, Intensive Care and Perioperative Medicine, Aisha bint Hamad Al Attiah Hospital, Hamad Medical Corporation, Doha P.O. Box 3050, Qatar; asaracoglu@hamad.qa; 3College of Medicine, Qatar University, Doha P.O. Box 2713, Qatar; 4Department of Anesthesiology, Intensive Care and Perioperative Medicine, Hazm Mebaireek General Hospital, Hamad Medical Corporation, Doha P.O. Box 3050, Qatar; 5Department of Anesthesiology and Intensive Therapy, Medical University of Lodz, 90-419 Lodz, Poland

**Keywords:** fluid responsiveness, geriatric, carotid ultrasound

## Abstract

Background: The reliability of determining fluid responsiveness during surgery in geriatric patients is challenging. Our primary outcome was to determine the reliability of Corrected Flow Time (FTc) in predicting fluid responsiveness. Methods: Elderly patients undergoing major surgery under general anesthesia were included. Measurements of common carotid artery diameter, velocity time integral, and systolic flow time (FT) were performed before and after a fluid challenge. FTc and carotid blood flow (CBF) were subsequently calculated. Results: The median change in carotid diameter was significantly higher in the fluid-responder (R) compared to the non-responder (NR) (6.51% vs. 0.65%, *p* = 0.049). The median change in CBF was notably higher in R compared to NR (30.04% vs. 9.72%, *p* = 0.024). Prior to the fluid challenge, systolic FT was significantly shorter in R than NR (285 ms vs. 315 ms, *p* = 0.027), but after the fluid challenge, these measurements became comparable among the groups. The change in systolic FT was higher in R (15.38% vs. 7.49%, *p* = 0.027). FTc and the change in FTc exhibited similarities among the groups at all study time points. Receiver operating characteristic analysis demonstrated an area under the curve of 0.682 (95% CI: 0.509–0.855, *p* = 0.039) for carotid diameter, 0.710 (95% CI: 0.547–0.872, *p* = 0.011) for CBF, 0.706 (95% CI: 0.540–0.872, *p* = 0.015) for systolic FT, and 0.580 (95% CI = 0.389–0.770, *p* = 0.413) for FTc. Conclusions: In geriatric patients, potential endothelial changes in the carotid artery may influence the dynamic markers of fluid responsiveness. Despite the demonstrated effectiveness of FTc in predicting fluid responsiveness in the general population, this study underscores the limited reliability of carotid Doppler ultrasonography indices for prediction in a geriatric patient population.

## 1. Introduction

The elderly patient demographic is experiencing a consistent increase, leading to a higher frequency of surgical procedures in clinical settings. Geriatric patients necessitate more vigilant monitoring and elevated levels of care, in contrast to their younger counterparts [1]. These patients often exhibit increased left ventricular diastolic dysfunction and decreased vascular reactivity, along with reduced venous compliance. As people age, the endothelial function of the blood vessels declines. Endothelial dysfunction leads to reduced vasodilation, impaired response to vasoactive substances, and increased vasoconstriction. This results in altered blood flow regulation, decreased tissue perfusion, and increased susceptibility to cardiovascular diseases such as hypertension and atherosclerosis. With aging, venous compliance decreases, leading to increased stiffness and a reduced ability to store blood. This can result in venous pooling, reduced venous return to the heart, and increased venous pressure. These changes highlight the importance of understanding age-related vascular changes in clinical practice and management [2]. Consequently, it becomes imperative to ascertain patients’ intravascular fluid status and responsiveness to fluid interventions since alterations in intercompartmental fluid exchange during surgery can result in exacerbated fluctuations in cardiac filling pressure [3]. Both hypovolemia and fluid overload can potentially have fatal consequences in this patient population. The term “fluid responsiveness” is defined as an increase in cardiac output (CO) of 10–15% in response to a fluid challenge [4]. Research has indicated that employing dynamic indices, which consider cardiac and pulmonary interactions rather than static methods, can lead to reduced mortality [5]. However, research on determining fluid responsiveness in geriatric patients is limited.

Traditionally, invasive methods like pulmonary artery catheters have been used for years to assess changes in cardiac output to assessing fluid responsiveness. More recently, minimally invasive measurement methods have emerged, incorporating dynamic indicators that account for cardiopulmonary interactions, such as stroke volume variation (SVV) and pulse pressure variation (PPV). Nevertheless, these approaches require the insertion of at least one arterial catheter. Studies have demonstrated that the Most Care^®^ monitor, which is supported by the Pressure Recording Analytical Method (PRAM) and does not require prior calibration, provides flow-based cardiac output measurement and vascular resistance measurement, and offers reliable results closely mirroring those obtained through right heart catheterization [6].

Recently, Doppler ultrasound of the common carotid artery has been used as a potential non-invasive tool to assess hemodynamic status [7]. Its advantages include its non-invasiveness, applicability regardless of patient position, ease of use due to the superficial location of the carotid artery, and independence from intrathoracic pressure changes. Carotid blood flow and corrected carotid flow time (FTc) provide information about systemic vascular resistance inversely proportional to the left ventricular preload. Additionally, the product of the velocity–time integral (VTI), which measures blood flow through the carotid artery during systole, and the cross-sectional area of the carotid artery allows for the determination of flow within the carotid artery. The reliability of determining fluid responsiveness with this method has been demonstrated in studies comparing it to invasive cardiac output measurements in the general adult patient population [8]. However, in the elderly population, where arterial elastance can be compromised and atherosclerotic changes are increased, the reliability of FTc in predicting fluid responsiveness may differ from that in the general patient population.

## 2. Objectives

We hypothesized that FTc, a carotid artery Doppler ultrasonography-derived parameter, may effectively predict fluid responsiveness in elderly patients as in the general population. The primary outcome of this study was to determine the reliability of FTc in predicting fluid responsiveness in elderly patients following a fluid challenge under general anesthesia. The secondary outcomes were to identify the cut-off point for FTc and the predictive ability of other carotid artery Doppler ultrasonography indices, such as carotid blood flow and systolic flow time, in fluid responsiveness.

## 3. Materials and Methods

### 3.1. Ethics

Ethical approval for this study (Ethical Committee No 09.2022.947) was provided by the Ethical Committee NAC of our University Hospitals, Istanbul, Turkey (Chairperson Prof H. Direskeneli) on 22 July 2022 and registered at ClinicalTrials.gov (NCT06087250).

### 3.2. Study Design

This prospective, observational study included patients aged 65 and above, classified as American Society of Anesthesiologists (ASA) I–III, undergoing major surgery that required invasive arterial line monitoring. The data were collected between July and December 2022 in operating theatres of our University Hospital. Patients with severe renal and cardiovascular diseases, fasting periods exceeding 8 h, or sepsis, and those who did not provide written consent, were excluded.

### 3.3. General Anesthesia Induction

If patients were not given premedication, they were taken to the operating room, and a 20G cannula was used to establish intravenous access. Routine monitoring included non-invasive blood pressure, ECG, SpO_2_, heart rate, temperature, BIS, and NMT. General anesthesia induction was achieved by titrating intravenous 1 mg kg^−1^ propofol, 1 mcg kg^−1^ fentanyl, and a bolus of 0.6 mg kg^−1^ rocuronium, followed by tracheal intubation once suitable conditions were ensured. After EtCO_2_ monitoring, volume-controlled mechanical ventilation was initiated with an 8 mL kg^−1^ tidal volume, 5 cmH_2_O PEEP, and respiratory rate was adjusted to maintain EtCO_2_ between 30 and 35 mmHg. An inspiratory-to-expiratory ratio (I:E) of 1:2 was used, with 50% oxygen and air. Anesthesia maintenance was achieved using remifentanil at 0.15–0.2 mcg kg^−1^ min^−1^ via intravenous infusion and 2% inhaled sevoflurane, adjusted according to the BIS target (40–60). An 18G second intravenous access was established for the fluid challenge.

### 3.4. Minimal Invasive Hemodynamic Monitoring (Most Care^®^)

Following anesthetic induction, a 20G intra-arterial catheter was placed in either the right or left radial artery. The transducer of the invasive arterial monitoring system was positioned at the 5th intercostal space along the anterior axillary line, parallel to the axillary line and zeroed. The Most Care^®^ device (Vygon, Écouen, France) was connected to the monitor via a Y cable. Hemodynamic variables, including mean arterial pressure (MAP), Cardiac Output (CO), and Stroke Volume Variation (SVV), were automatically calculated and recorded by the device. 

### 3.5. Carotid Ultrasonography

Common Cartotid Doppler ultrasound was performed by an experienced anesthetist trained in this area using a 50 mm linear array, 12–5 mHz ultrasound probe (Philips, Sparq USG systems, Amsterdam, The Netherlands). To image the common carotid, a linear probe was placed horizontally at the lower border of the patient’s thyroid cartilage in the neck. After obtaining an image of the carotid, the probe was rotated longitudinally to obtain a B-mode image along the long axis of the carotid. Using calipers, the diameter was measured approximately 2 cm proximal from the carotid bifurcation. The cross-sectional area (CSA) of the carotid was calculated using the formula πr^2^. By placing a pulse wave (PW) and adjusting the angle to be less than 60 degrees, blood flow was visualized. The time between the start point, where the Doppler flow was directed upward, and the beginning of the next cardiac cycle time was measured. The time from the starting point of the upward-directed Doppler flow to the dicrotic notch was measured (Figure 1). Flow time was divided by the square root of the cycle time to calculate the corrected carotid blood flow time (FTc) using Bazett’s formula (FTc = ST √CT^−1^) [9]. The velocity–time integral (VTI) was calculated for the flow imaged with PW. Carotid blood flow (CBF) was calculated using the following formula: CSA × VTI × heart rate. The examiner conducted three consecutive measurements, and the means of these values were used for analysis. 

Measurements were performed when the patient was stable after tracheal intubation, with a criterion of a less than 10% change in mean arterial pressure over three minutes. Baseline minimal invasive hemodynamic monitoring using the MostCare^®^ device, and dynamic indices values were recorded in hemodynamically stable patients. Baseline measurements for carotid diameter, carotid blood flow, carotid VTI, systolic flow time, cycle time, and corrected flow time were performed simultaneously with carotid Doppler ultrasound (T_1_). Subsequently, to assess fluid responsiveness, a 6 mL kg^−1^ dose of intravenous balanced crystalloid solution was administered within 30 min by one researcher (BB). Measurements with MostCare^®^ and carotid Doppler were repeated (T_2_), and patients were divided into two groups based on fluid responsiveness. Patients were categorized into responders (R) and non-responders (NR), with a 10% increase in cardiac output, as measured by another researcher, defining fluid responders (AS).

### 3.6. Sample Size Calculation

In the study conducted by Kimura et al., the area under the ROC curve for the corrected carotid blood flow velocity in predicting fluid responsiveness was found to be 0.82 [1]. In our study, based on previous research and assuming a fluid-responsive/fluid-unresponsive ratio of 1, we calculated the sample size with an area under the curve of 0.75, a type 1 error of 0.05, and a power of 0.90 [10]. A total of 40 patients are required for the study. Anticipating a 10% patient data loss, the study aims to include 44 patients.

### 3.7. Statistical Analysis

The R (project for statistical computing), version 2.15.3 program was used for statistical analysis. Data were reported using minimum, maximum, mean, standard deviation, median, first quartile, third quartile, frequency, and percentage. The normality of quantitative data was assessed using the Shapiro–Wilk test and graphical inspection. The Mann–Whitney U test was used for intergroup comparisons of quantitative variables, and the Wilcoxon signed-ranks test was employed for intragroup comparisons. Qualitative data were compared using Pearson chi-square test, Fisher’s exact test, and Fisher–Freeman–Halton exact test. ROC analysis was used to determine predictive values. The cut-off point was chosen based on the Youden Index. ROC curves were compared using the DeLong method. The grey zone approach, as described by Coste and Pouchot, was applied to determine an inconclusive range of carotid measurement values [11]. The cut-off values defining the grey zone were established based on a sensitivity and specificity of 90%. Statistical significance was set at *p* < 0.05.

## 4. Results

In this study, 55 patients were assessed for eligibility; 15 were excluded due to end-stage renal failure (*n* = 9), carotid stenosis (*n* = 5), and sepsis (*n* = 1). Thus, 40 patients were enrolled and included in the analysis. Patient characteristics were comparable between non-responders (*n* = 22) and responders (*n* = 18, Table 1).

Non-responders (NR) had a significantly higher MAP before the fluid challenge as compared to responders (R) (87.5 mmHg vs. 73 mmHg, *p* = 0.049). After the fluid challenge, MAP was similar among the groups. While there was a statistically significant difference among the groups, there was no clinical significance for MAP values. HR was comparable before and after the fluid challenge between the study groups. Responders had a significantly higher SVV before the fluid challenge as compared to non-responders (14.5% vs. 8.5%, *p* = 0.004); no significant difference remained after the fluid challenge (9% vs. 8%, *p* = 0.388, respectively).

Carotid diameter remained comparable before and after the fluid challenge across the study groups. The carotid diameter significantly increased after the fluid challenge compared to before the fluid challenge in the responder group. The median change in carotid diameter was significantly higher in R compared to NR (6.5% vs. 0.6%, *p* = 0.049). The carotid VTI significantly increased in both responders (18.7%, *p* = 0.001) and non-responders (13.2%, *p* = 0.026) after the fluid challenge. The carotid VTI and ∆ carotid VTI were similar among the study groups. CBF exhibited comparability across the respective groups both before and after the fluid challenge. The median change in CBF (∆CBF) was significantly greater in responders compared to non-responders (30% vs. 9.7%, *p* = 0.024). Prior to the fluid challenge, responders exhibited a significantly shorter systolic flow time (285 ms) compared to non-responders (315 ms), with a *p*-value of 0.027. After the fluid challenge, no statistically significant difference persisted in systolic FT. Nevertheless, both responders and non-responders experienced a significant increase in systolic FT after the fluid challenge, with a Δ systolic FT of 15.3% (*p* < 0.001) for responders and 7.4% (*p* = 0.011) for non-responders. Responders demonstrated a significantly greater Δ systolic FT increase compared to non-responders (*p* = 0.027). After the administration of fluid, there was a significant increase in FTc in responders (9.8%, *p* = 0.049) and non-responders (5.8%, *p* = 0.049); however, there was no statistically significant difference between the groups (Table 2).

The ability of changes in carotid ultrasound Doppler indices to predict fluid responsiveness is shown in Table 3. For ∆ carotid diameter (%), the optimal cut-off value is >1.2, with a sensitivity of 83.3% and specificity of 59.0%, and AuROC curve is 0.6 (95% CI = 0.509, 0.855, *p* = 0.039). The optimal cut-off value is >5.4 for ∆ CBF (%), with a sensitivity of 88.8% and specificity of 50%, and AuROC curve is 0.7 (95% CI = 0.547, 0.872, *p* = 0.011). For ∆ systolic FT (%) the optimal cut-off value is >11.3, with a sensitivity of 66.6% and specificity of 68.1%, and AuROC curve is 0.7 (95% CI = 0.540, 0.872, *p* = 0.015). The optimal cut-off value is >19 for ∆ FTc (%), with a sensitivity of 33.3% and specificity of 95.4%, and AuROC curve is 0.5 (95% CI = 0.389, 0.770, *p* = 0.413).

A grey-zone strategy is utilized to determine the thresholds for alterations in carotid Doppler ultrasonography indices while maintaining a fixed sensitivity or specificity of 90% to assess fluid responsiveness. The grey zone for ∆ carotid diameter (%) occurs between −6 and 12.3%, containing 32 (80%) patients. The grey zone for ∆ CBF (%) occurs between −7 and 53.5, containing 24 (60%) patients. The grey zone for ∆ systolic FT (%) occurs between 0 and 20, containing 24 (60%) patients. The grey zone for ∆ FTc (%) occurs between −6.7 and 18.9, containing 29 (72.5%) patients. The comparison of AuROC curves using the DeLong method demonstrated no statistically significant differences among the investigated carotid Doppler ultrasonography indices (Table 4).

## 5. Discussion

This study evaluated fluid responsiveness using carotid Doppler measurements in geriatric patients undergoing major surgery. The validity of the method was confirmed using Mostcare parameters [6]. Among the indices obtained through carotid Doppler ultrasonography, changes in FTc values were found to be similar between responders and non-responders after the fluid challenge. Significant changes were observed in carotid diameter, carotid blood flow, and systolic FT after the fluid challenge in responder patients. Systolic FT was significantly shorter in the responder group before the fluid challenge, which became similar to that of the non-responder group after fluid replacement. Although the changes were statistically significant before and after the fluid challenge in both groups, the increase in systolic FT was significantly higher in the fluid-responsive group compared to the non-responsive group within the defined grey zone. However, it was observed that 60% of patients were within the grey zone. Similarly, for carotid blood flow, 60% of patients were found to be within the defined grey zone, and for carotid diameter, 80% of patients were within the grey zone. This indicates that a significant proportion of patients fall within the indeterminate or grey zone for systolic FT, carotid blood flow, and carotid diameter. These findings emphasize the challenges and potential limitations of relying solely on these parameters to assess fluid responsiveness in geriatric patients undergoing major surgery.

The assessment of fluid responsiveness using carotid ultrasonography is gaining popularity. The ease of use, affordability, minimal training requirements, and non-invasive nature of carotid ultrasound contribute to its attractiveness [12]. Carotid blood flow and carotid-corrected flow time measurements serve as indicators of cardiac output. Flow time, a corrected mechanical systole time derived from the Doppler spectrogram, represents the direct relationship between heart rate and systolic duration, providing an estimation of cardiac output [13,14]. FTc, when the carotid diameter remains constant, corresponds to the time it takes for the ejected volume to pass through the carotid during systole. An increase in this value is expected with a higher stroke volume, particularly in fluid-responsive patients. The FTc can be calculated using different formulas. FTc values calculated with different formulas may demonstrate varying levels of correlation and concordance in reflecting cardiac output (CO) changes. Among these formulas, Bazzett’s equation has been shown to obtain the highest performance in depicting the relationship between FTc and CO [9]. Therefore, we utilized Bazzett’s equation in our study. 

In both groups of our study, following a fluid challenge, FTc values exhibited similar changes. Possible reasons for this similarity include the development of benign sinus arrhythmia or heart valve changes associated with aging without causing significant major restrictions. Although such changes may not be diagnosable by ultrasound examination in our patients, it is speculated that these potential alterations could have affected carotid traces and, consequently, rendered the measurement of the dicrotic notch in ultrasound inconclusive. Ma et al. demonstrated, in their study, that pulmonary artery catheter measurements did not yield significant results for FTc, mirroring our findings [12]. The noteworthy prevalence of atrial fibrillation in almost half of the patients in this study significantly influenced the lack of significance of the FTc outcomes. Similarly, in a study conducted in patients with septic shock, FTc values did not correlate with increases in mean arterial pressure [15]. This suggests that, in patient cohorts with potential endothelial damage, the utility of FTc in indicating fluid responsiveness remains a subject of debate. 

Kim et al. [16]. demonstrated that, in patients undergoing brain surgery, pre-anesthetic induction FTc predicted fluid responsiveness with a cutoff value of 349.4 ms (AuROC = 0.8). However, the mean age of patients included in this study was 54 (21–76 years), encompassing both younger and older individuals. In a study involving elderly patients undergoing abdominal surgery under general anesthesia, they showed that the performance of diagnosing fluid responsiveness using a cut-off value of 340.74 ms for FTc was satisfactory (AuROC = 0.8, 95% CI = 0.721–0.900, *p* < 0.01); however, approximately 50% of the patients fell within the grey zone [17]. To predict post-induction hypotension in elderly patients, an FTc cut-off value of 379.1 ms was found to be reliable, with 25.3% of patients falling within the grey zone [18]. The dissimilarity between post-induction hypotension and fluid responsiveness as concepts suggests that the predictive capability of FTc for post-induction hypotension may differ from its ability to predict fluid responsiveness.

In our study, carotid blood flow increased by 30% in responders after the fluid challenge, while no significant change was observed in non-responders. The optimal threshold for the change in carotid blood flow (∆CBF > 5.4%) was determined, with an AuROC level of 0.7. Sixty percent of patients fell within the grey zone defined for this value. The greater impact of age-related changes in blood flow in the carotid arteries on elderly patients compared to the general population may suggest that this value may not be highly useful in predicting fluid responsiveness. Studies have demonstrated a decrease in the ratio or volume flow of carotid blood flow with increasing age in age-matched controls [19]. According to the flow-mediated dilation phenomenon, an increase in blood flow leads to an increase in shear stress, which, in turn, enhances nitric oxide (NO) synthase activity, resulting in vasodilation [20]. According to this phenomenon, there should have been an increase in carotid diameter in both groups. A significant increase in carotid diameter was only observed in the fluid-responsive group following the fluid challenge, which suggests that carotid diameter measurement may be a dynamic parameter.

An increase in systolic FT after the fluid challenge was observed in both groups. In the responder group, an almost twofold change was observed in response to fluid compared to the non-responder group, suggesting that this could be considered as a surrogate. However, when considering the cut-off value, the low sensitivity and high specificity of systolic FT, along with 60% of cases remaining in the grey zone, indicate that this alone may be insufficient in determining fluid responsiveness. The finding in our study that FTc lacks predictive ability for fluid responsiveness supports the notion that systolic FT alone may also be insufficient.

This study aimed to determine fluid responsiveness and guide the titration of intravenous fluids, ensuring optimal fluid balance and minimizing the risk of complications associated with inadequate or excessive fluid administration, which all have an impact on clinical outcome [4]. This study underscores the reliability of the carotid artery Doppler ultrasound indices in predicting fluid responsiveness in geriatric patients, which limits its clinical utility. Currently, our data do not support the usage of these indices alone, or even when considered only as adjuncts in clinical decision-making at the bedside. Additionally, future studies should evaluate the ability of carotid Doppler parameters to predict fluid responsiveness in a larger geriatric patient population while also investigating their influence on clinical outcomes.

## 6. Limitations

There are several limitations to our study. The first limitation is that it is a single-center study. The second limitation is that ultrasound measurements were performed by a single individual. Due to the manual calculation of Carotid VTI measurements, there is a margin of error. Although the number of patients in the study was appropriate according to the previously calculated sample size, the high number of patients remaining in the gray zone may have caused limitations.

## 7. Conclusions

In conclusion, assessing fluid responsiveness in geriatric patients undergoing surgery proves to be challenging when compared to younger counterparts. The stiffness of the vascular structure, potential endothelial changes, and the dynamic markers of an aged cardiovascular structure are believed to influence this evaluation. Despite the proven effectiveness of FTc in predicting fluid responsiveness in the general adult population, this study highlights the limited reliability of the assessed carotid Doppler ultrasonography indices for predicting fluid responsiveness in the geriatric patient population. FTc poorly discriminated fluid responders from non-responders.

The following information was previously known:The precise determination of fluid responsiveness in geriatric patients is challenging.The dynamic markers of fluid responsiveness are affected in geriatrics because of the aged cardiovascular structures and potential endothelial changes.

Our study obtained the following new results:Doppler ultrasound parameters of carotid diameter, carotid blood flow, and systolic flow time after a fluid challenge were evaluated in responder patients.Despite the demonstrated effectiveness of the corrected flow time in predicting fluid responsiveness in the general population, this study underscores its limited reliability in geriatric patients.

## Figures and Tables

**Figure 1 healthcare-12-00783-f001:**
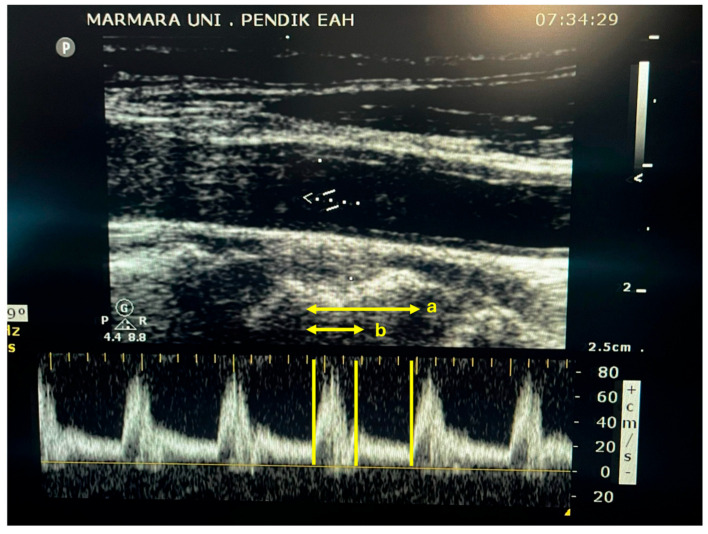
Carotid Doppler ultrasound image, showing a pulsed-wave Doppler signal. a: total cycle time; b: systolic flow time.

**Table 1 healthcare-12-00783-t001:** Clinical characteristics of patients.

	Non-Responder (*n* = 22)	Responder (*n* = 18)	Total (*n* = 40)	*p*
Age (years)	72 (67, 75)	67.5 (67, 73)	70 (67, 74)	^a^ 0.305
BSA (m^2^)	1.95 (1.77, 2)	1.95 (1.83, 2)	1.95 (1.8, 2)	^a^ 0.713
Gender				^b^ 0.949
Female	10 (45.5)	8 (44.4)	18 (45)	
Male	12 (54.5)	10 (55.6)	22 (55)	
ASA Score				^c^ 0.092
I	3 (13.6)	1 (5.6)	4 (10)	
II	11 (50)	15 (83.3)	26 (65)	
III	8 (36.4)	2 (11.1)	10 (25)	
Comorbidities	20 (90.9)	17 (94.4)	37 (92.5)	^d^ 0.999
Chronic Hypertension (>140/90 mmHg)	16 (72.7)	16 (88.9)	32 (80)	^d^ 0.258
Diabetes Mellitus	5 (22.7)	5 (27.8)	10 (25)	^d^ 0.731
Coronary Heart Disease	7 (31.8)	3 (16.7)	10 (25)	^d^ 0.464
COPD	0 (0)	1 (5.6)	1 (2.5)	^d^ 0.450

^a^ Mann–Whitney U test, reported as median (first quartile, third quartile), ^b^ Pearson chi-square test, ^c^ Fisher–Freeman–Halton exact test, ^d^ Fisher’s exact test, BSA: body surface area, ASA: American Society of Anesthesiology, COPD: Chronic Obstructive Pulmonary Disease.

**Table 2 healthcare-12-00783-t002:** Hemodynamic variables and carotid doppler ultrasonography indices.

		Non-Responder (*n* = 22)	Responder (*n* = 18)	^a^ *p*
MAP (mmHg)	Before	87.5 (70, 104)	73 (65, 84)	0.049 *
	After	84 (74, 91)	83 (75, 93)	0.828
	Change (%)	1.4 (−20, 12.7)	17 (1.41, 27)	0.015 *
	^d^ *p* (Change)	0.375	0.006 *	
HR (bpm)	Before	73 (68, 82)	77 (70, 83)	0.376
	After	69 (61, 78)	75 (68, 83)	0.211
	Change (%)	−6.4 (−10.9, −1.2)	−3.5 (−8.5, 1.2)	0.399
	^d^ *p* (Change)	0.006 *	0.112	
SVV (%)	Before	8.5 (6, 13)	14.5 (10, 18)	0.004 *
	After	8 (4, 11)	9 (8, 11)	0.388
	Change (%)	−8.8 (−33.3, 28.5)	−35.7 (−47, 11.1)	0.174
	^d^ *p* (Change)	0.326	0.038 *	
Carotid diameter (mm)	Before	7.7 (7.2, 8.2)	7.3 (6.3, 8.1)	0.120
	After	7.8 (7.3, 8.3)	7.8 (6.7, 8.6)	0.653
	Change (%)	0.6 (−1.3, 2.9)	6.5 (1.5, 8.5)	0.049 *
	^d^ *p* (Change)	0.407	0.003 *	
Carotid VTI (cm)	Before	17.9 (14.6, 25.9)	18.2 (14, 19.5)	0.703
	After	21.2 (17.4, 25)	21 (19.9, 24.9)	0.817
	Change (%)	13.2 (1.1, 22.7)	18.7 (11.1, 53.7)	0.211
	^d^ *p* (Change)	0.026 *	0.001*	
CBF (mL/min)	Before	21,793.9 (13,666.3, 29,597.4)	26,371.7 (12,420, 45,758.5)	0.514
	After	20,732.7 (15,160.1, 28,859.1)	29,021.5 (20,826, 70,737.9)	0.069
	Change (%)	9.7 (−12.4, 26.3)	30 (13.9, 56.4)	0.024 *
	^d^ *p* (Change)	0.322	<0.001 *	
Systolic FT (ms)	Before	315 (294, 360)	285 (264, 306)	0.027 *
	After	339 (306, 360)	335.5 (324, 360)	0.881
	Change (%)	7.4 (−1.6, 14.8)	15.3 (4.1, 32.6)	0.027 *
	^d^ *p* (Change)	0.011 *	<0.001 *	
FTc (ms)	Before	342.5 (323, 360)	345.5 (313, 389)	0.946
	After	356.5 (337, 385)	367 (347, 400)	0.226
	Change (%)	5.8 (0, 11.18)	9.8 (−2.27, 25)	0.392
	^d^ *p* (Change)	0.049 *	0.049 *	

Reported as median (first quartile, third quartile), ^a^ Mann–Whitney U test, ^d^ Wilcoxon signed-ranks test, * *p* < 0.05. MAP: mean arterial pressure, HR: heart rate, SVV: stroke volume variation, VTI: velocity time integral, CBF: carotid blood flow, FT: flow time, FTc: corrected flow time.

**Table 3 healthcare-12-00783-t003:** Prediction of fluid responsiveness by ROC curves of changes in carotid ultrasound doppler indices.

90% (Sensitivity or Specifity)	AuROC (95% CI)	*p*	Cut-Off	Grey Zone	Patients in Grey Zone (%)	Sensitivity (95% CI)	Specificity (95% CI)	Youden Index
∆Carotid diameter (%)	0.6 (0.5–0.8)	0.039 *	>1.2	−6–12.3	32 (80)	83.3 (58.6–96.4)	59 (36.4–79.3)	0.4
∆CBF (%)	0.7 (0.5–0.8)	0.011 *	>5.4	−7–53.5	24 (60)	88.8 (65.3–98.6)	50 (28.2–71.8)	0.3
∆Systolic FT (%)	0.7 (0.5–0.8)	0.015 *	>11.3	0–20	24 (60)	66.6 (41.0–86.7)	68.1 (45.1–86.1)	0.3
∆ FTc (%)	0.5 (0.3–0.7)	0.413	>19	−6.7–18.9	29 (72.5)	33.3 (13.3–59.0)	95.4 (77.2–99.9)	0.2

ROC: receiver operating characteristics; CBF: carotid blood flow; FT: flow time; FTc: corrected flow time. * *p* < 0.05.

**Table 4 healthcare-12-00783-t004:** Comparing ROC curves with the DeLong method.

	Carotid Diameter	Carotid BF	Systolic FT
CBF	0.792	-	-
Systolic FT	0.828	0.972	-
FTc	0.371	0.299	0.022 *

CBF: blood flow; FT: flow time; FTc: corrected flow time. * *p* < 0.05.

## Data Availability

Data are contained within the article.

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
