# Peer review of "The Reliability of Carotid Artery Doppler Ultrasonography Indices in Predicting Fluid Responsiveness during Surgery for Geriatric Patients: A Prospective, Observational Study"

_healthcare, 2024, doi:10.3390/healthcare12070783_

Round 1

Reviewer 1 Report

Comments and Suggestions for Authors

Comments on the Quality of English Language

Reviewer 2 Report

Comments and Suggestions for Authors

1.    Is it required by your journal? Ethical Committee NAC of our University Hospitals, Istanbul, Turkey (Chairperson Prof H. Direskeneli)

2.    lower border of the patient's thyroid in the neck may pl be further clarified whether its gland or cartilage?.

3.    Mann-Whitney U test was used for intergroup comparisons of quantitative variables, and the Wilcoxon signed-ranks test was employed for intragroup comparisons.

a.    These are cause-and-effect relationships which require regression tests or corelation tests. Pl justify use of the tests

Reviewer 3 Report

Comments and Suggestions for Authors

Thank you for the invitation to review the manuscript by Dr. Bılgılı, et al., “The Reliability of Carotid Artery Doppler Ultrasonography Indices in Predicting Fluid Responsiveness for Geriatric Patients: a prospective, observational study.” The authors describe the utility of ultrasound-measured corrected carotid blood flow time (FTc) in the carotid artery in patients undergoing “major surgery” as a surrogate of increasing cardiac output after an intravenous fluid challenge of 6 mL/kg by at least 10% or not, as measured by an noninvasive, proprietary device. Eighteen (45%) of patients were “responders”. Numerous other ultrasound variables were measured and compared, many of which intuitively changed with volume infusion. The authors found an optimal cut-off value of >19.077 for ∆FTc (%), with a sensitivity of 33.33%, specificity 95.45% and ROC 0.580. There was no significant difference between groups in this parameter, but there were in others. 

The manuscript is well written, legible, and with few grammatical errors. The introduction is logical, organized, and well cited. The anesthesia provided was well described and standardized. The methods were well described, including the statistical analysis. They included 40 patients in the analysis (of a calculated sample size of 44), raising the possibility of underpowering the analysis. This study deserves commendation for adding to the literature on carotid Doppler assessment. 

  • In table 1, the majority of patients had “normal” blood pressure and heart rate. In this setting, I would expect that most of the patients would not benefit from volume loading regardless of the test performed. Since 80% of patients fell within a grey zone, I would again suspect the population studied has a low prevalence of “hypovolemia”. 
  • The test cut-off (delta FTc > 19.1%) had a very high specificity (95.5%) and therefore has a low false-positive rate. Perhaps there may be value in this test in a broader population. However, it seems the “standard” measurement of SVV held true in this study and the analysis of this variable deserves comment. 

Minor:

  • in the abstract, “R” and “NR” need to be defined. 
  • please comment on the sample size in limitations. 

Reviewer 4 Report

Comments and Suggestions for Authors

INTRODUCTION

- 3rd sentence needs a reference. 

- Providing brief explanations or definitions for terms such as "vascular reactivity" and "venous compliance" could enhance understanding for a broader audience. 

METHODS

-       Please provide rational/justification behind certain statistical tests used in the analysis.

DISCUSSION

- Provide a more in-depth discussion on the limitations and real world clinical impact based on this study.

- A future directions component of the discussion specifically highlighting clinical utility and its effects on improving patient monitoring and outcomes is necessary.

Comments on the Quality of English Language

Moderate editing of the English language. 

Author Response

Please see the attachment. The revised manuscript with the changes marked red according to the comments is attached below.
